# The Impact of Sociodemographic, Macroeconomic, and Health Status and Resources on Infant Mortality Rates in Oman: Evidence from 1980 to 2022

Rawaa Abubakr Abuelgassim Eltayib [ID], Mohammed Al-Azri [ID] and Moon Fai Chan *[ID]

Department of Family Medicine and Public Health, College of Medicine and Health Sciences, Sultan Qaboos University, Muscat P.O. Box 123, Oman; s136163@student.squ.edu.om (R.A.A.E.); mhalazri@squ.edu.om (M.A.-A.)
* Correspondence: moonf@squ.edu.om; Tel.: +968-2414-1132

**Abstract:** Background: The infant mortality rate (IMR) is an important reflection of the well-being of infants and the overall health of the population. This study aims to examine the macroeconomic (ME), sociodemographic (SD), and health status and resources (HSR) effects on IMR, as well as how they may interact with each other. Methods: A retrospective time-series study using yearly data for Oman from 1980 to 2022. Partial Least Squares-Structural Equation Modelling (PLS-SEM) was utilized to develop the exploratory model of the determinants of IMR. Results: The model indicates that HSR determinants directly, but negatively, affect IMR ($\beta = -0.617$, $p < 0.001$). SD directly and positively affects IMR ($\beta = 0.447$, $p < 0.001$). ME only indirectly affects IMR ($\beta = -0.854$, $p < 0.001$). ME determinants also exert some direct influences on both HSR ($\beta = 0.722$, $p < 0.001$) and SD ($\beta = -0.916$, $p < 0.001$) determinants. Conclusions: This study has indicated that IMR is a multi-faceted issue. It also highlighted the interplay of numerous variables and their influence on IMR, particularly the role that social status, the health sector, and the wealth of the country and its population play in reducing IMR. These findings indicate that an integrated policy that addresses socioeconomic and health-related factors and the overall ME environment is necessary for the health and well-being of the children and the population overall in Oman.

**Keywords:** infant mortality rate; macroeconomic; sociodemographic; health status and resources; Oman; partial least squares structural equation model

## 1. Introduction

A key health indicator to which much attention is paid is the infant mortality rate (IMR). The World Health Organization (WHO) defines IMR as "The probability of a child born in a specific year or period dying before reaching the age of one, if subject to age-specific mortality rates of that period" [1]. IMR is expressed as a rate with the numerator being the number of fatalities among children under one year of age during a particular time, over the denominator of the total number of live births during the same time, expressed as per 1000 live births [2]. IMR is an important indicator of child survival because it provides insight into the social and economic milieu in which children (and other members of society) live and the healthcare services they obtain [3].

Countries worldwide started paying more attention to health indicators in 2000 when a Millennium Summit was held to encourage the attending countries to create a world in which maintaining progress and ending poverty would be a top priority. One of the health-related indicators chosen to measure this goal was the IMR [4]. According to recent statistics, infants remain an important sociodemographic on which we ought to concentrate our efforts, with a United Nations Inter-Agency Group for Child Mortality Estimation (UN IGME) report indicating that before the age of five, more than 5 million children passed away in 2021 [5,6]. Over half of those deaths, 2.4 million, were newborn deaths. Though the IMR trend worldwide has been declining steadily, almost 10 thousand children under

the age of one died every day in 2020, compared to 24,000 in 1990 [7]. Oman is a country located on the Arabian Peninsula's south-eastern coast. It is also a proud member of the Gulf Cooperation Council (GCC), an association of six nations that serves as a regional, intergovernmental, political, and economic entity. Since the 20th century, the GCC nations' economies have experienced remarkable growth, thus helping to reduce the IMR in the region substantially [8]. In Oman, the earliest recorded IMR was very high. It stood at 118 infant deaths per 1000 live births in 1970. However, a dramatic decline occurred in mortality rates just before the start of the 20th century, reaching 16.7 deaths per 1000 live births. This downward trend continued steadily until it reached about 8.1 infant deaths per 1000 live births in 2021 [9]. Even though Oman has made tremendous progress in lowering IMR over the past few years, additional steps must be taken to reduce it further, particularly given that, despite experiencing comparable economic conditions, Oman's IMR remains the highest in the GCC area.

Identifying the factors which have a massive influence on IMR is the first step in tackling this issue. IMR is often influenced by several determinants documented in several papers. These factors can broadly be divided into three categories: Health status and resources (HSR), macroeconomic (ME), and Sociodemographic (SD) determinants. Inequalities in familial income and other social-based policies are critical in controlling infant outcomes. This association exists in many countries with different cultural and economic identities. For example, USA and Western Europe [10], Brazil [11], and Nepal [12]. The wealthier the family, the easier for their infants to receive healthcare, even if they need an expensive procedure or medication [13]. Total and adolescent fertility rates were also documented to impact children's survival chances worldwide [14,15].

Various articles have shed light on the effect of HSR variables on IMR. For instance, health status variables relating to the infant are critical to that infant's survival chances. Prematurity and its complication, low birth weight, and sepsis are all documented factors affecting IMR worldwide negatively [16]. A well-rounded healthcare system is essential for maintaining the well-being of a community. However, good health facilities not being well-utilized by their intended population can harm their health and increase infant death chances [17]. One of the metrics used to gauge access to healthcare services and the effectiveness of their coverage is the universal health coverage (UHC) index. The Global Health Observatory describes it as a composite index expressed on a unitless scale from 0 to 100, and the score is calculated based on several indicators. This index is meant to reflect the coverage of some essential health services, notably maternal and child health provisions [18]. Achieving UHC has been associated with overall improved health outcomes—including reduced morbidity and mortality for all ages, particularly children—for countries [19]. Access to quality health services is not only important during the antenatal period, but is also essential during labor itself. Birth attendance by skilled health professionals has been documented to increase the chances of infant survival [20]. Their presence is essential for minimizing complications during labor and delivery and providing time-sensitive interventions that significantly reduce the risks of morbidity and mortality for the mother and her infant [21]. Another substantial factor that the literature has highlighted is immunization coverage. Vaccinations are deemed one of the best interventions ever introduced, considering how many lives are saved cost-effectively. Since the implementation of immunization programs, the world has managed to avert about 3.5 to 5 million children's deaths annually [22].

Worldwide, the impact of ME factors on IMR has been discussed in various articles, usually highlighting the impact of gross domestic product per capita (GDPpc) increase on reducing IMR [15,23]. An ecological study has indicated the presence of this phenomenon in 83 low-income and middle-income countries [24]. In the GCC region, and as a result of the significant expansion of their economy and its performance as assessed by their gross domestic product (GDP), both overall and per capita, the GCC has significantly improved the overall health of their communities, as well as significantly reduced infant mortality compared to the past [7,25,26]. This demonstrates the significant impact of GDP and, by

extension, macro-economic factors overall on IMR. GDPpc was also an important factor in reducing mortality rates in Oman [27]. To better address the regional determinants of IMR, a systematic review and meta-analysis were undertaken in the GCC area [28]. It highlighted several factors that were linked with the likelihood of infant deaths. Interestingly, numerous extracted variables aligned with international ones, whereas others were unique to the Arabian Peninsula.

In Oman, some studies have investigated IMR risk factors. However, most of them seem to be focused on investigating a certain factor's effect on IMR [29,30] or have undergone an investigation on a singular hospital only [31–35]. Overall, there is a scarcity of research that endeavored to investigate concurrently how macroeconomic (ME), sociodemographic (SD), and health status and resources (HSR) determinants influenced the IMR in Oman. This study will attempt to achieve this aim by creating an explanatory model incorporating all the factors mentioned earlier that affect IMR in Oman. This model will also try to evaluate potential interactions between these factors with six objectives:

1. To explore the effects of ME on IMR in Oman;
2. To explore the effects of SD on IMR in Oman;
3. To explore the effects of HSR on IMR in Oman;
4. To explore the effects of ME on HSR in Oman;
5. To explore the effects of ME on SD in Oman;
6. To explore the effects of SD on HSR in Oman.

## 2. Materials and Methods

### 2.1. Study Design

This retrospective time-series design uses annual data from 1980 to 2022 in Oman. The 1980s and after were chosen for the study period because there were not many reliable government records from Oman available before that time.

### 2.2. Data Resources, Data Collection, and Sample Size

All the data that were used in this study were secondary. All the variables and their descriptions were gathered from international, open sources, and some local and regional sources. Local sources were mainly the Omani National Centre for Statistics and Information [36] and GCC-stat [37]. The World Bank [38], OIC Statistics (OICStat) Database by SESRIC [39], and The Global Health Data Exchange (GHDx) data catalog by Institute for Health Metrics and Evaluation (IHME) [40] were the three international open data sources utilized for this project.

This study's required sample size (the number of years) was determined based on the number of indicators pointing at any determinant in the proposed model or a minimum $R^2$ at 0.25. The required years ranged from 30 to 59 years, with approximately 80 percent power at a 5 percent significance level [41–43].

To ensure the level of relevance and viability of these data, it was necessary to set up evaluation criteria. These criteria included assessing: the methods used to collect the data, the accurate assessment by comparing data from different sources, the time between collections to ensure that it was still valid for our research problem, the definition of dependent and explanatory variables, and the units of measurement and categories used. Additionally, a data cleaning procedure was undertaken and was considered one of the steps conducted to ensure the data's quality. Principally, imputation procedures will only be attempted on the missing random explanatory variables [44]. Furthermore, any item in this study that was found to have 20% missing data or more will be excluded and is considered ineligible to be entered into the analysis procedure entirely. In this study, multiple regression imputation was the primary technique to complete the dataset's gaps [45].

### 2.3. The Proposed Conceptual Model

The proposed conceptual model was developed based on previous studies [43–46] and the results from a systematic review [29]. Overall, this study intends to examine the effect of multiple explanatory variables on IMR in Oman using structural equation modeling (SEM). In the model, there are both Latent and Manifest variables. Latent variables (LV), or constructs, are not directly measurable, while Manifest variables (MV) can be measured directly. In this study, the four LVs (HSR, SD, ME, and IMR) incorporate multiple associated MVs that try to reflect that specific LV accurately. There are six hypotheses formulated to address the six objectives of the study. For a more detailed illustration of the indicators and theorized relationships, see Figure 1.

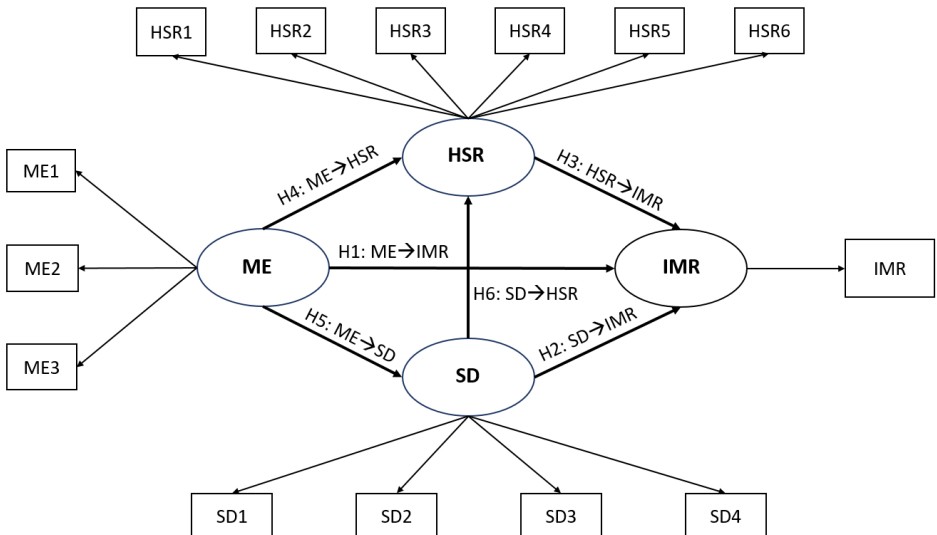

**Figure 1.** Hypothetical Conceptual Model of the Study. Ellipse: Latent variables; Rectangle: Manifest variables; IMR: Infant Mortality Rate, SD: Sociodemographic, HSR: Health status and resources; ME: Macroeconomic; ME1: GDP pc; ME2: Employment to population ratio (15+, males); ME3: Employment to population ratio (15+, females); SD1: Birth rate (crude); SD2: Adolescent fertility rate; SD3: Fertility rate (total); SD4: Percentages of women married; Newborns protected against tetanus (%); HSR2: One-year-olds immunized with BCG; HSR3: One-year-olds immunized with MCV1; HSR4: One-year-olds immunized with Pol3; HSR5: Births attended by skilled health personnel; HSR6: UHC service coverage index. Six hypotheses in the conceptual model (H1: ME→IMR; H2: SD→IMR; H3: HSR→IMR; H4: ME→HSR; H5: ME→SD; H6: SD→HSR).

A systematic review and meta-analysis were first undertaken to assess all the available literature in the GCC region, including Oman, to understand what factors affected IMR in the area [28]. Consequently, the selected MVs for this research are guided by the findings of this review. For a more detailed view of every variable used in the study and their descriptions and definitions, refer to Table 1. The proposed variables to reflect the HSR construct were: Vaccination coverage as measured by four indicators (percentage of newborns protected against tetanus, as well as the percentage of 1-year-olds immunized with BCG, MCV1, and Pol3), Births attended by skilled health personnel, and universal health coverage (UHC) service coverage index [47–49]. To reflect the SD construct, the following MVs were suggested: crude birth rate, adolescent fertility rate, total fertility rate, and percentages of women married [25,27,50]. To reflect the ME construct, the following MVs were used: GDPpc, and employment to population ratio [26,47,51].

**Table 1.** The list of all variables included in the analysis procedure and their definitions.

| Label | Variable | Definition | Source |
|---|---|---|---|
| IMR | Infant Mortality Rate | The infant mortality rate is the number of infants dying before one year per 1000 live births yearly. | World Bank |
| SD variables | | | |
| SD1 | Birth rate (crude) | Crude birth rate indicates the number of live births per 1000 midyear population. | World Bank |
| SD2 | Adolescent fertility rate | Adolescent fertility rate is the number of births per 1000 women aged 15–19. | World Bank |
| SD3 | Fertility rate (total, births per woman) | Total fertility rate represents the number of children that would be born to a woman if she were to live to the end of her childbearing years and bear children in accordance with age-specific fertility rates of the specified year | World Bank |
| SD4 | Percentages (%) of women married | Proportion of married or in-union women of reproductive-age (ages 15–49) in total population of women in the same age group, expressed in terms of percentage. | SESRIC |
| HSR variables | | | |
| HSR1 | Newborns protected against tetanus (%) | Percentage of births by women of child-bearing age who are immunized against tetanus, i.e., newborns who achieved protection at birth from their mothers who received tetanus toxoid doses during the pregnancy. | World Bank |
| HSR2 | One-year-olds immunized with BCG (%) | One-year-olds who have received one dose of Bacille Calmette-Guérin (BCG) vaccine in a given year, expressed in terms of percentage. | SESRIC |
| HSR3 | One-year-olds immunized with MCV1 (%) | One-year-olds who have received at least one dose of measles-containing vaccine (MCV1) in a given year, expressed in terms of percentage. | SESRIC |
| HSR4 | One-year-olds immunized with Pol3 (%) | One-year-olds who have received three doses of polio vaccine (Pol3) in a given year, expressed in terms of percentage | SESRIC |
| HSR5 | Births attended by skilled health personnel (%) | Birth attended by Skilled Health worker in a given period (as % of total birth) | IHME-GHDx |
| HSR6 | UHC service coverage index (0 to 100) | Coverage of essential health services | IHME-GHDx |
| ME variables | | | |
| ME1 | GDP per capita (current US$) | GDP per capita is gross domestic product divided by midyear population | World Bank |
| ME2 | Employment to population ratio (15+, males) in % | Proportion of a country's female population (with ages 15+) that is employed, expressed in terms of percentage. | World Bank |
| ME3 | Employment to population ratio (15+, females) in % | Proportion of a country's male population (with ages 15+) that is employed, expressed in terms of percentage. | World Bank |

SESRIC: Statistical, Economic and Social Research and Training Centre for Islamic Countries; IHME-GHDx: Institute for Health Metrics and Evaluation-Global Health Data Exchange; ME: Macroeconomic; SD: Sociodemographic; HSR: Health Status and resources.

### 2.4. Statistical Analysis

Researchers are gradually utilizing SEM, a second-generation technique, to circumvent first-generation procedures' constraints [52]. In SEM, covariance-based SEM (CB-SEM) and Partial least-square SEM (PLS-SEM) are available to estimate relationships between LVs and MVs. CB-SEM technique is used mostly to either affirm or debunk hypotheses [53]. At the same time, PLS-SEM is chiefly concerned with helping to develop or create theories in exploratory-focused studies [54,55] and has been used on other health indicators, e.g., life expectancy, in Bahrain [45]. This exploratory study intends to discover which explanatory variables significantly affect IMR in Oman and investigate how they may interact with each other. Henceforth, PLS-SEM is the better choice of modeling techniques to try and materialize these relationships.

PLS-SEM results are reviewed and evaluated systematically to determine how well the model fits the data. Firstly, the measurement models are the primary focus of PLS-SEM model evaluation. Some reliability and validity measures are needed to examine the measurement quality [56]. Reliability tests include Cronbach's Alpha (CA), Composite Reliability (CR), Rho Alpha (Rho-A), and MV's loading. Reliability tests will be conducted to take a score above 0.70 [57]. In order to assess validity, various tests will be used, such as the Average Variance Extracted (AVE), as well as Hetero-Trait and Mono-Trait 2 (HTMT2). Multiple studies [57–61] advise having a score above 0.50 for the AVE test, while for the HTMT2 test, a score below 1.0 is suggested. Secondly, if the measurement characteristics of constructs are acceptable, the structural model estimates are next in the systemic process of evaluating PLS-SEM results. To achieve that, there are multiple metrics to evaluate the structural model and indicate the model's predictive capabilities. For example, the coefficient of determination ($R^2$), and the size and statistical significance

of the structural path coefficients. These two serve as the main evaluation criteria for PLS-SEM outcomes. The predictive relevance ($Q^2$) and effect sizes ($f^2$) provide further information regarding the quality of the PLS path model estimations [62]. As mentioned earlier, some recommended value cut-off points involve a score of more than 0.25 for the $R^2$, which signifies high levels of explained variance. A beta (β) coefficient higher than 0.20 is advised; otherwise, such relationships are indicated to be removed from the model [63,64]. Numerous studies [53,60,65] suggest classifying the predictive relevance ($Q^2$) into three categories: strong (>0.35), moderate (0.02–0.35), or weak (<0.02). Effect size can be determined by calculating Cohen's $f^2$. The general guidelines for assessing $f^2$ effect size dictate that if a relationship with direct effect has values less than 0.02, then it is a weak relationship or has no effect. Values between 0.02 and 0.35 indicate a moderate relationship, while values above 0.35 indicate a strong effect [63,64,66]. The bootstrapping procedure was used to assess the significance of path coefficients. This study's weighting iteration was set at 300 with a bootstrapping of 10,000 subsamples, as recommended by several studies [64,67]. The IBM SPSS v21 and SmartPLS v3 software programs will be used to perform the above analyses, and all significant levels were set at $p < 0.05$.

## 3. Results

### 3.1. Characteristics of the Variables in the Final Model

There were three MVs chosen to represent the ME construct: GDPpc, employment to population ratio (15+, males), and employment to population ratio (15+, females). To reflect the SD construct, the total fertility rate and percentages of women married were chosen in the final model. Finally, to portray the HSR construct, the percentage of newborns protected against tetanus and the percentage of 1-year-olds immunized with Tuberculosis (BCG), at least one dose of measles (MCV1), and three doses of polio (Pol3) vaccines, were used. Table 2 summarizes the descriptive statistics of the variables chosen for the final model.

**Table 2.** Descriptive statistics of the variables in the final model.

| Label | N | Min | Max | Mean | SD | Skewness | Kurtosis |
|-------|---|-----|-----|------|-----|----------|----------|
| IMR | 43 | 9.50 | 78.90 | 24.09 | 19.53 | 1.44 | 1.08 |
| SD3 | 43 | 2.57 | 8.13 | 4.64 | 2.013 | 0.64 | −1.24 |
| SD4 | 43 | 55.21 | 69.47 | 61.884 | 5.89 | 0.28 | −1.76 |
| HSR1 | 43 | 29.00 | 99.00 | 83.77 | 21.22 | −1.84 | 1.94 |
| HSR2 | 43 | 51.00 | 99.00 | 94.07 | 10.74 | −3.14 | 9.69 |
| HSR3 | 43 | 10.00 | 99.00 | 88.26 | 23.30 | −2.41 | 4.66 |
| HSR4 | 43 | 18.00 | 99.00 | 88.51 | 23.62 | −2.31 | 3.92 |
| ME1 | 43 | 5073.85 | 24,722.64 | 11,899.81 | 6770.15 | 0.66 | −1.20 |
| ME2 | 43 | 43.88 | 87.80 | 73.93 | 11.06 | −1.19 | 1.34 |
| ME3 | 43 | 16.07 | 27.59 | 22.16 | 3.18 | −0.07 | −1.17 |

IMR: Infant Mortality Rate, SD3: Fertility rate; SD4: % of woman married; HSR1: Tetanus immunization (%); HSR2: Bacille Calmette–Guérin (BCG) immunization (%); HSR3: Measles-containing-vaccine first-dose (MCV1) immunization (%); HSR4: Polio vaccine (Pol3) immunizations (%); ME1: Gross domestic product per capita (GDP pc); ME2: Employment to population ratio of males (15+) in %; ME3: Employment to population ratio of females (15+) in %; Min.: Minimum; Max.: Maximum; SD.: Standard deviation; N: Number of years.

### 3.2. The Final Model Evaluation Indices

The first step in evaluating the final model is to review its outer model measurements. This is undertaken firstly by reviewing the four reliability tests. The results of the reliability tests are summarized in Table 3. These findings demonstrate that the reliability of the final model is firmly established since all of the reliability tests' scores were above 0.7. Next, the model's validity measures are then examined. The findings suggest that the final model has achieved convergent and discriminant validity, with all AVE measures scoring above 0.7. The HTMT ratio values are below 1 and abide by the recommendation that its 95% confidence interval lies between +1 and −1 (Table 3).

**Table 3.** Reliability, validity, and predictive capabilities of The Final Model.

| LV | MV | FL | CA | Rho-A | CR | $R^2$ | $Q^2$ | AVE | HTMT (95% CI) | | | |
| --- | --- | --- | --- | --- | --- | --- | --- | --- | --- | --- | --- | --- |
| | | | | | | | | | IMR | ME | SD | HSR |
| IMR | IMR | 1.000 | 1.000 | 1.000 | 1.000 | 0.959 | 0.942 | 1.000 | - | - | - | 0.934 (0.854–0.970) |
| ME | ME1 | 0.893 | 0.938 | 0.963 | 0.960 | - | - | 0.889 | 0.851 (0.763–0.910) | - | - | 0.728 (0.606–0.818) |
| | ME2 | 0.946 | | | | | | | | | | |
| | ME3 | 0.987 | | | | | | | | | | |
| SD | SD3 | 0.990 | 0.978 | 0.981 | 0.989 | 0.839 | 0.816 | 0.978 | 0.878 (0.833–0.928) | 0.954 (0.913–0.983) | - | 0.698 (0.595–0.815) |
| | SD4 | 0.988 | | | | | | | | | | |
| HSR | HSR1 | 0.944 | 0.978 | 0.979 | 0.984 | 0.521 | 0.474 | 0.940 | - | - | - | - |
| | HSR2 | 0.953 | | | | | | | | | | |
| | HSR3 | 0.992 | | | | | | | | | | |
| | HSR4 | 0.988 | | | | | | | | | | |

LV: Latent variable; MV: Manifest variables; FL: Factor Loadings; CA: Cronbach's Alpha; Rho-A: Rho-Alpha; CR: composite reliability; AVE: Average variance extracted; HTMT: Hetero-Trait and Mono-Trait ratio; $R^2$: coefficient of determination; $Q^2$: Predictive relevance; ME: Macroeconomic; ME1: GDP pc; ME2: Employment to population ratio (15+, males); ME3: Employment to population ratio (15+, females); SD: Sociodemographic; SD3: Fertility rate (total); SD4: Percentages of women married; HSR: Health status and resources; HSR1: Newborns protected against tetanus (%); HSR2: One-year-olds immunized with BCG; HSR3: One-year-olds immunized with MCV1; HSR4: One-year-olds immunized with Pol3; A result above 0.7 is considered satisfactory for all the reliability tests (FL, CA, Rho-A, CR); A score above 0.50 is advised for the AVE test, while for the HTMT test, a score below 1.0 is suggested, or an HTMT ratio with a confidence interval between −1 and +1 is satisfactory to establish discriminant validity; For $R^2$: a score of more than 0.25 is recommended. For $Q^2$: the values are categorized into strong (>0.35), moderate (0.02–0.35), or weak (<0.02) predictability.

Evaluation of the model's prediction and explanatory capabilities is the next step. Regarding the explanatory power of this model, $R^2$ values explain about 95.9% of the variance in IMR, 52.1% of the variance in HSR, and 83.9% of the variance in SD. Additionally, the $Q^2$ values show that this model can achieve strong predictability, because the $Q^2$ values ranged from 0.474 for HSR, 0.816 for SD, and 0.942 for IMR. $Q^2$ estimates above 0.35 indicate a powerful predictive capability of a model (Table 3). A graphical representation of the final model is presented in Figure 2.

This final model validates hypothesis 2 and shows that SD exerts a positive and strong direct effect on IMR (H2: SD→IMR), with a β coefficient of 0.447 ($p < 0.001$) and an effect size of 4.891. The final model substantiates hypothesis 3, as it suggests that HSR has a strong yet negative direct influence on IMR (β = −0.617, $f^2$ = 4.891, $p < 0.001$) (H3: HSR→IMR). Additionally, the final model indicates that the relationship between ME and HSR exists (H4: ME→HSR) and is a strong, positive, and direct relationship with β equal to 0.722 ($p < 0.001$). Notably, this final model improved this pathway's effect size, indicating a moderate relationship to a $f^2$ value of 1.087. Moreover, hypothesis 5 (H5: ME→SD) shows that ME exerts a strong but negative impact on SD, with a β coefficient of −0.916 ($p < 0.001$) and a $f^2$ value of 5.193. Finally, hypothesis 1 (H1: ME→IMR) and hypothesis 6 (H6: SD→HSR) were not significant in the final model. For an overview of the effect and effect sizes of the final model, refer to Table 4.

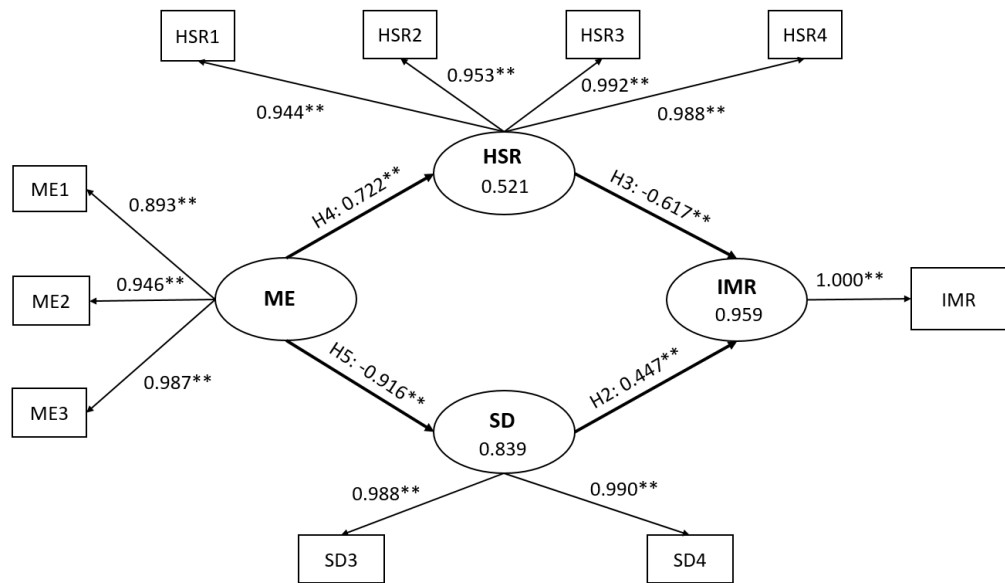

**Figure 2.** The Final Model of the Study. IMR: Infant Mortality Rate; SD: Socio-demographic; HSR: Health status and resources; ME: Macroeconomic; ME1: GDP pc; ME2: Employment to population ratio (15+, males); ME2: Employment to population ratio (15+, females); SD3: Fertility rate (total); SD4: Percentages of women married; HSR1: Newborns protected against tetanus (%); HSR2: One-year-olds immunized with BCG; HSR3: One-year-olds immunized with MCV1; HSR4: One-year-olds immunized with Pol3; H2: Hypothesis 2; H3: Hypothesis 3, H4: Hypothesis 4; H5: Hypothesis 5; **: indicate significant values at $p < 0.001$.

**Table 4.** The effects and effect sizes of the four latent variables (ME, SD, HSR, and IMR) in the Final Model.

| Hypothesis | Pathway | Direct Effect (95% CI) | Indirect Effect (95% CI) | Total Effect (95% CI) | $f^2$ |
|---|---|---|---|---|---|
| H1 | ME→IMR | - | −0.854 (−0.898 to −0.790) | −0.854 (−0.898 to −0.790) | - |
| H2 | SD→IMR | 0.447 (0.327 to 0.601) | - | 0.447 (0.327 to 0.601) | 2.560 |
| H3 | HSR→IMR | −0.617 (−0.712 to −0.458) | - | −0.617 (−0.712 to −0.458) | 4.891 |
| H4 | ME→HSR | 0.722 (0.605 to 0.806) | - | 0.722 (0.605 to 0.806) | 1.087 |
| H5 | ME→SD | −0.916 (−0.945 to −0.880) | - | −0.916 (−0.945 to −0.880) | 5.193 |
| H6 | SD→HSR | - | - | - | - |

Hypotheses: H1, H2, H3, H4, H5, and H6; HSR: Health status and resources, IMR: Infant mortality rate, ME: Macroeconomic, SD: Sociodemographic, $f^2$: effect size. For $f^2$: >0.35 indicates a strong effect, 0.02–0.35 indicates a moderate effect, and <0.02 indicates a weak or no relationship. All effects are positive at $p < 0.001$.

## 4. Discussion

### 4.1. The Effects of SD Determinants on IMR

Based on the final model of this study, SD has a direct, positive, and strong effect on IMR. Furthermore, the SD determinant in Oman is best reflected by two MVs: The total fertility rate and the percentage of married women of reproductive age (15–49). Having the total fertility rate as an indicator of the infant's survival chances in this study is consistent with findings from other publications. For instance, in Malaysia, the fertility rate was considered the primary determining factor of IMR and a contributing variable in the Philippines and Thailand [68]. Locally, a study investigated the effects of multiple factors on IMR in Oman using regression analysis. Their findings also indicated that the fertility rate is a key predictor of IMR in Oman [27]. A high fertility rate suggests that the mother will give birth more frequently, and as the number of births rises, so do the chances that a complication may develop during the pregnancy. Obstetric problems are typically associated with a greater chance of long-term medical issues, leaving the mother in poor health. [69]. This ill health during pregnancy will dramatically increase the risk of morbidity and mortality for both the mother and her child [69–71]. The final model has

highlighted the percentage of married women of reproductive age (aged 15–49 years) as an SD indicator. This factor's trend has been decreasing in Oman, from 69% in 1980 to 57% in 2020 [39]. Its reduction is speculated to result from increased mean age at marriage in Oman. Due to the indirect effects of this factor on fertility and birth rates, it has contributed to affecting IMR in this country.

### 4.2. The Effects of HSR Determinants on IMR

In Oman, HSR determinants exert a direct but negative impact on IMR, similar to the findings of other studies [72,73]. Furthermore, the final model indicated that the HSR construct in Oman is best reflected by four MVs: percentage of newborns protected against tetanus, and percentage of 1-year-olds immunized with BCG, MCV1, and Pol3. Immunization programs have proven to be some of the most efficient and economical public health initiatives ever implemented by nations. Vaccines help mitigate about 3.5–5 million fatalities yearly from infections such as measles, diphtheria, tetanus, pertussis, and influenza [22]. Higher vaccination rates are a reflection of the effectiveness and quality of a nation's healthcare system, as well as how well the populace makes use of it. The higher and more thorough the vaccination coverage is, the lower the IMR in a country. This finding was similarly demonstrated in other studies [74,75]. The Expanded Program on Immunization (EPI) initiative was introduced in 1981 in Oman. Following that, a thorough childcare program was implemented nationwide in 1987, along with the introduction of the child health card and child health register (MR2 register). Consequently, vaccination coverage rose throughout the latter half of the 1980s and the beginning of the 1990s, and it has since maintained a steady level of around 98–99% [76]. It is speculated that Oman has seen a significant decrease in the diseases encompassed by immunization measures due to implementing all these policies [77].

### 4.3. The Effects of ME Determinants on IMR

This study's model highlighted an indirect and negative influence between ME factors and IMR in Oman ($\beta = -0.854$, $p < 0.001$). This finding was consistent with the results of other publications [78,79]. Moreover, the final model indicated that the ME construct in Oman is best reflected by three MVs: GDPpc, employment to population ratio of 15+ aged males, and employment to population ratio of 15+ aged females. This study's finding of GDPpc association with IMR is consistent with the results of other studies worldwide [14] and in Oman [27]. GDPpc reflects the standard of living of the population of a particular country. Therefore, its improvement points toward comfortable living conditions in the community and has been linked to better health outcomes overall, particularly lower chances of infants dying [80]. Another indication of a country's economic situation is its population's employment proportion. Several studies have highlighted the important role of the higher employed proportion of the population in reducing mortality rates [11]. Interestingly, this research found that the employment status of both genders is associated with IMR in Oman. This finding is the opposite of the GCC region systematic review, which noted that maternal employment is insignificant in affecting IMR and that only paternal working status matters [28]. Employed parents can provide better living conditions for their children and better quality of care. Furthermore, the parents' working status is associated with better healthcare-seeking behavior and utilization of healthcare facilities [11,13].

### 4.4. The Effects of ME Determinants on HSR

This research also highlighted some interactions between the determinants themselves. One of those interactions is the direct and positive impact ME determinants exert on the HSR determinants ($\beta = 0.722$, $p < 0.001$). This finding was similar to other studies [81,82]. Notably, this connection is thought to exist due to how the country's healthcare spending is impacted by its financial situation. When a nation's economy is booming, there is more space in the health sector's budget and more money available to fund the application and expansion of health programs [83].

### 4.5. The Effects of ME Determinants on SD

This study has indicated that ME determinants have a direct and negative impact on SD factors in Oman ($\beta$ = −0.916, $p$ < 0.001). This was consistent with the findings of other studies [43,48]. A population with well-educated, employed mothers and a flourishing economy will exhibit lower fertility rates. A study exploring the determinants of fertility rates in 21 Sub-Saharan African countries indicated that the country's financial conditions are essential factors influencing fertility rates' decline [84]. The wealthier a country and its population are, the more drastic the decline in fertility rates was observed. Higher educational attainment and a greater standard living standard may raise people's awareness of and usage of family planning choices, which is a likely explanation for this effect [85]. Some theories from some articles point toward the effect of financial circumstances on marriages. Furstenberg notes the effect of economic determinants on the mean age of marriage. He observed that with the rising financial situation of both the country and, in turn, the individuals themselves, females tended to delay their marriages [86]. This phenomenon is also theorized to exist locally. Oman has experienced an economic boom since the start of the 20th century, and an increase has followed in the employment proportion of its male and female populations. This has been speculated to influence the reduction in marriage rates and the rise in the mean age of marriages in the country [87].

### 4.6. Strengths and Limitations of This Study

This was the first study that considered ME, SD, and HSR determinants of IMR in Oman, from both a country-level and a multi-factorial perspective. Some findings of this research were consistent with the results of other studies conducted in Oman previously, while other findings (such as the employment status of females) were in contrast with those studies. However, this research should be considered a stepping-stone for future studies. It would also be interesting to attempt to recreate a PLS-SEM model for other countries to determine IMR by including some of the variables excluded from this study due to missing data limitations. It would be especially intriguing to fully comprehend the impact of environmental factors and educational and healthcare expenditure on the model and how they influence IMR. Another limitation of this research is related to its nature. Since this was an ecological study, it is subject to ecological fallacy. Ecological fallacies occur when, for example, after group-level data have been gathered and analyzed, associations at the level of the individual are presumed to be affected by the findings. In other words, conclusions about the individuals cannot be extrapolated from results based on these far larger population-level results. However, aggregate data studies should be considered an important starting point for research in a country. Overall, the finding of this research project should be treated as an indicator of the overall situation of IMR from a multi-factorial perspective. Moreover, this research's findings should prompt future studies to try and give proper attention to key variables highlighted by this study.

## 5. Conclusions

The final model of this study has indicated the substantial effects that HSR determinants exert on IMR in Oman. Particularly, the key factors that reflected the HSR construct were the immunization coverage of measles, tuberculosis, and polio and the percentage of newborns with at-birth protection from tetanus. These variables reflect the critical role of the health sector in reducing IMR. Before the proper implementation of the immunization programs, all diseases, as mentioned earlier, contributed to significant morbidity and mortality in the children of Oman. Hence, this signifies the essential job of these intervention programs in achieving lower mortality rates and overall population well-being. This study has noted the effect of ME determinants on reducing IMR in Oman. However, this research notes that ME does not affect IMR directly, but indirectly by influencing both HSR factors and SD determinants. This suggests that more care should be taken to not only keep the nation's economy strong overall but also to try and concentrate on where the money ought to be spent. Generally, translating a better economic situation into reducing and eliminating

health inequalities should be at the forefront of governmental strategy. Political support for this strategy has been documented to control health-related inequalities significantly and effectively [88]. Furthermore, some strategies should be implemented to cushion the fallouts related to financial downturns, particularly in the health sector.

**Author Contributions:** Conceptualization, M.F.C.; methodology, M.F.C. and R.A.A.E.; formal analysis, R.A.A.E.; data curation, M.F.C. and R.A.A.E.; writing—original draft preparation, M.F.C. and R.A.A.E.; writing—review and editing, M.F.C., R.A.A.E. and M.A.-A.; supervision, M.F.C. and M.A.-A.; funding acquisition, M.F.C. and M.A.-A. All authors have read and agreed to the published version of the manuscript.

**Funding:** This research was funded by Sultan Qaboos University Internal Grant: IG/MED/ FMCO/22/02.

**Institutional Review Board Statement:** The study was conducted in accordance with the Declaration of Helsinki and obtained an exemption approved by the Medical Ethics Committee of Sultan Qaboos University (MREC #2654l).

**Informed Consent Statement:** Not applicable.

**Data Availability Statement:** Data supporting reported results can be found online.

**Acknowledgments:** The authors would like to acknowledge the comments from the reviewers.

**Conflicts of Interest:** The authors declare no conflict of interest.

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
