# Peer review of "The Impact of Sociodemographic, Macroeconomic, and Health Status and Resources on Infant Mortality Rates in Oman: Evidence from 1980 to 2022"

_ejihpe, doi:10.3390/ejihpe13060075_

Round 1

Reviewer 1 Report

This study is meaningful in that it provides evidence for policy making by identifying the factors that affect Oman's IMR. In the introduction, the need for research was specifically presented, and there is no problem with the composition and system of the thesis. It is also meaningful in that it guided the direction of follow-up research by specifically presenting the strengths and limitations of this study.

- Please describe in the discussion section what differentiates the results of this study from those of previous studies. 

- And please describe this by emphasizing it in the strengths and weaknesses section of your study.

- In Figure 1, H4: ME-IMR is written, but please modify IMR to HSR.

- Please edit references according to the author guidelines for references.

Author Response

This study is meaningful in that it provides evidence for policy making by identifying the factors that affect Oman's IMR. In the introduction, the need for research was specifically presented, and there is no problem with the composition and system of the thesis. It is also meaningful in that it guided the direction of follow-up research by specifically presenting the strengths and limitations of this study.

Response:

Thank you very much for taking the time to review the manuscript. We deeply appreciate your support and kind words.

- Please describe in the discussion section what differentiates the results of this study from those of previous studies.

Response:

Thank you for your comment. Initially, we have a paragraph in the introduction section (lines 110-115) highlighting the major differences between our study and others previously undertaken in the area – though it is mainly focused on methodological and design differences-. Furthermore, some mention of similarities and differences between our results and the findings of another research done in Oman was discussed in the discussion section (lines 281-283 and lines 329-331). In addition, some findings were compared with other studies worldwide that investigated infant mortality rate determinants (lines 325-329). We added a clarification in the discussion section, where necessary, further highlighting the differences between the findings of this study from previous ones conducted in Oman.

- And please describe this by emphasizing it in the strengths and weaknesses section of your study.

Response:

We appreciate your input. We have added a few points in the strengths and limitation sub-heading.

- In Figure 1, H4: ME-IMR is written, but please modify IMR to HSR.

Response:

Thank you for your comment. Yes, we amended it in Figure 1. It should be H4: ME-HSR.

- Please edit references according to the author guidelines for references.

Response:

Your comment is appreciated. The references were further edited to correspond with the author guidelines for MDPI Journals.

Reviewer 2 Report

I consider the work with minor revisions publishable. I invite the authors to stress the fact that IMR is dependent on wealth (both in the abstract and in the results) and that women's work role is dependent on social status

Author Response

I consider the work with minor revisions publishable. I invite the authors to stress that IMR is dependent on wealth (both in the abstract and in the results) and that women's work role is dependent on social status.

Response:

Thank you very much for your comment and input. We appreciate it greatly. Firstly, as per your recommendations, the role of wealth in affecting IMR has been added to the abstract section.

Secondly, regarding underlining the role that wealth plays in influencing IMR, though it is not stated explicitly, in the results and discussion section, we detail how the country's and population's economic situation may affect IMR. In particular,

The results section mentions the economy’s indirect role as a determinant of IMR in Oman (lines 268-270). Furthermore, in the discussion section, we mention how the macroeconomic variables reflect the population's standard of living (lines 320-325). It is mentioned that the more comfortable the living situation is, the lesser the mortality outcomes are.

Regarding women’s working status being affected by social status. Though this point is valid and very important, it is not easy to discuss it within the scope of our model. This point suggests that the females’ social status (which can fall under the Sociodemographic determinant -SD-) can affect their employment status (which falls under the Macroeconomic determinant -ME-). This suggests a directional relationship in the shape of SD→ME. Unfortunately, our model did not take such relationships into account. Thus, in this article, we only focused on employment (and other macroeconomic factors) effects on the social situation of females (as per the 5th hypothesis of the model: ME→SD).